# The Law of Entropy Increase and the Meissner Effect

**DOI:** 10.3390/e24010083

**Published:** 2022-01-03

**Authors:** Alexey Nikulov

**Affiliations:** Institute of Microelectronics Technology and High Purity Materials, Russian Academy of Sciences, Moscow District, 142432 Chernogolovka, Russia; nikulov@iptm.ru

**Keywords:** law of entropy increase, irreversible thermodynamic process, generation of Joule heat, reversible thermodynamic process, phase transition, superconducting transition, Meissner effect, assumption of molecular disorder, persistent current

## Abstract

The law of entropy increase postulates the existence of irreversible processes in physics: the total entropy of an isolated system can increase, but cannot decrease. The annihilation of an electric current in normal metal with the generation of Joule heat because of a non-zero resistance is a well-known example of an irreversible process. The persistent current, an undamped electric current observed in a superconductor, annihilates after the transition into the normal state. Therefore, this transition was considered as an irreversible thermodynamic process before 1933. However, if this transition is irreversible, then the Meissner effect discovered in 1933 is experimental evidence of a process reverse to the irreversible process. Belief in the law of entropy increase forced physicists to change their understanding of the superconducting transition, which is considered a phase transition after 1933. This change has resulted to the internal inconsistency of the conventional theory of superconductivity, which is created within the framework of reversible thermodynamics, but predicts Joule heating. The persistent current annihilates after the transition into the normal state with the generation of Joule heat and reappears during the return to the superconducting state according to this theory and contrary to the law of entropy increase. The success of the conventional theory of superconductivity forces us to consider the validity of belief in the law of entropy increase.

## 1. Introduction

The theory of superconductivity [1,2] is an outstanding achievement of twentieth-century physics. The famous Ginzburg–Landau (GL) theory [1] published in 1950 describes superconductivity as a macroscopic quantum phenomenon with the help of the GL wave function. The GL theory [1] is based on the assumption that numerous electrons can be on the same level. It is impossible according to the basis of quantum mechanics since electrons are fermions rather than bosons. The main efforts of the theorists dealing with superconductivity in the forties and fifties were aimed at explaining how electrons can turn into bosons. According to the majority of physicists, this was done in 1957 by J. Bardeen, L.N. Cooper, and J.R. Schrieffer (BCS) [2]. According to the BCS theory, electrons become bosons due to the formation of Cooper pairs of electrons [2].

The conventional theory of superconductivity [1,2] successfully describes numerous quantum phenomena observed in superconductors. The Meissner effect, discovered experimentally in 1933 [3], is described as a special case of the quantization of the magnetic flux, discovered experimentally in 1961 [4,5], and the the quantization of the persistent current, discovered experimentally in 1962 [6]. One of the most outstanding achievements of the theory is the description of the Abrikosov state [7] observed in type II superconductors [8]. The achievements of the conventional theory of superconductivity [1,2] are so great that, until recently, no one questioned this theory.

However, recently Jorge Hirsch drew attention to the internal inconsistency of this theory: on the one hand, this theory was created within the framework of equilibrium reversible thermodynamics, and on the other hand, it predicts Joule heating [9,10,11]. The inconsistency is obvious since Joule heating is an irreversible thermodynamic process that cannot be described in the framework of equilibrium reversible thermodynamics. Therefore, it is strange that no one noticed this inconsistency earlier. The inconsistency is considered in the paper [12] as a consequence of the history of superconductivity investigation and the belief of most scientists in the law of entropy increase.

The transition of a bulk superconductor into the normal state in a magnetic field *H* was considered as an irreversible process before the discovery the Meissner effect [3]. Expert in superconductivity D. Shoenberg wrote in the book [13] published in 1952: “*At that time* [before 1933], *it was assumed that the transition in a magnetic field is substantially irreversible, since the superconductor was considered as a perfect conductor (in the sense which was discussed in Chapter II), in which, when the superconductivity is destroyed, the surface currents associated with the field are damped with the generation of Joule heat*”.

However, if this transition is an irreversible thermodynamic process during which the total entropy increases on $\Delta S=E_{k}/T$ due to the dissipation of the kinetic energy $E_{k}$ of the electric current into Joule heat, then the Meissner effect is experimental evidence of a process reverse to the irreversible thermodynamic process during which Joule heat is converted back into the kinetic energy and the total entropy decreases on $\Delta S=-E_{k}/T$. Belief in the law of entropy increase forced physicists to change their understanding of the superconducting transition, which began to be considered as a phase transition after 1933. Therefore, the conventional theory of superconductivity [1,2] was created within the framework of equilibrium reversible thermodynamics.

This theory explains the Meissner effect [3], the quantization of the magnetic flux [4,5] and of the persistent current [6] as a consequence of the appearance of the persistent current after the transition into the superconducting state because of the quantization of the angular momentum of Cooper pairs. The persistent current is annihilated after the transition into the normal state. No generation of Joule heat should be at this transition in order to be considered as the phase transition. Therefore, the famous physicist W. H. Keesom wrote in 1934 that “*it is essential that the persistent currents have been annihilated before the material gets resistance, so that no Joule-heat is developed*” [14].

However, the conventional theory of superconductivity [1,2] does not explain how the persistent currents can be annihilated before the material gets resistance. This theory considers the process of disappearance of the persistent current during the transition into the normal state in the same way as this process was considered before 1933: the persistent currents are damped with the generation of Joule heat. This internal inconsistency, indicated for the first time in [9,10,11], means that the conventional theory of superconductivity [1,2] contradicts the law of entropy increase since the appearance of the persistent current after the transition into the superconducting state is the process reverse to the irreversible thermodynamic process according to this theory.

The success of the conventional theory of superconductivity [1,2] in the description of numerous phenomena observed in superconductors prompts us to consider the validity of the almost universal belief in the law of entropy increase. The history and the basis of this belief will be considered in the Section 2. In the Section 3, reader’s attention will be drawn to the change of the sense of the law of entropy increase after the victory of the atomistic-kinetic worldview over the thermodynamic-energy worldview. The assumption of molecular disorder needed for the validity of the law of entropy increase and examples of its violation in quantum systems will be considered in the Section 4.

Jorge Hirsch states that the conventional BCS theory of superconductivity [2] is wrong because of the prediction of Joule heating [9,10,11]. He ignores numerous experimental evidences of Joule heating observed in superconductors that corroborate this prediction of [1,2]. Hirsch also ignores the obvious mistakes that physicists had to make after the discovery of the Meissner effect in order to maintain their faith in the law of entropy increase. These mistakes will be considered in the Section 5. The urgency of an unbiased and open discussion of the possibility of violating the law of entropy increase is justified in the Section 6.

## 2. The History of the Law of Entropy Increase

The concept of entropy as a physical property was established for thermodynamic systems by the second law of thermodynamics. This concept came from German scientist Rudolf Clausius who formulated in the 1850s the Carnot principle [15] with the help of this concept. Therefore as Smoluchowski wrote in 1914: “*We call the Carnot principle as the second law of thermodynamics since Clausius’s time*” [16]. Sadi Carnot postulated in 1824 that the efficiency of conversion of heat into work with the help a heat engine has the upper limit
(1)$$\eta_{max}=1-\frac{T_{co}}{T_{he}},\;$$
which is determined by the ratio $T_{co}/T_{he}$ of the cooler temperature $T_{co}$ to the heater temperature $T_{he}$. In particular, the efficiency of any heat engine is zero η=0 without a temperature difference $T_{he}-T_{co}$, according to the Carnot Equation (Equation 1). Therefore, the temperature difference $T_{he}-T_{co}$ must be created and maintained in order the heat energy ΔQ can be converted in an useful work *W*.

The efficiency factor of any heat engine is equal by definition to the part $\eta =W/Q_{he}$ of the thermal energy of the heater $Q_{he}$ that is converted into useful work *W*. This part has a maximum value when all other parts of the thermal energy created in the heater is spent only on increasing the termal energy of the cooler, i.e., $W=Q_{he}-Q_{co}$. Therefore, the maximum efficiency should be equal $\eta_{max}=1-Q_{co}/Q_{he}$. This equation corresponds to the Carnot Equation (Equation 1) when the value $Q_{co}/T_{co}-Q_{he}/T_{he}$ cannot be less than zero. The inequality $Q_{co}/T_{co}-Q_{he}/T_{he}\geq 0$ expresses the essence the first rigorous definition of the second law of thermodynamics proposed by Clausius in the 1850s: heat can never pass from a colder to a warmer body without some other change, connected therewith, occurring at the same time. This definition was based on the concept of entropy as the ratio S=Q/T of the thermal energy *Q* of a thermodynamic body to its thermodynamic temperature *T*. The entropy of a closed system cannot decrease, i.e., the law of entropy increase is another formulation of the second law of thermodynamics.

### 2.1. The Irreversibility Is Needed for the Impossibility of a Perpetuum Mobile

Sadi Carnot substantiated his principle by the centuries-old belief in the impossibility of perpetuum mobile. He wrote in his brilliant work of 1824 [15] that “*it would be not only a perpetual motion, but also an unlimited creation of the driving force without the cost of phlogiston or any other agents*” if the efficiency of a heat engine could exceed the maximum value (1). The genius of Carnot is that using the notion of phlogiston, he determined the relationship between maximum efficiency of any heat engine and the impossibility of a perpetuum mobile, which is valid in different ideas about heat.

It became clear that Carnot postulated irreversibility in physics when the heat began to consider as a form of energy. The law of entropy increase postulates most clearly the irreversibility of some thermodynamic processes. The car stops when it runs out of fuel, because the kinetic energy $E_{k}$ of its directional movement is converted into thermal energy $Q=E_{k}$. The total entropy increases on the value Q/T in this process since the kinetic energy of a directed motion of any macroscopic body does not contribute to the entropy. This process is irreversible according to the law of entropy increase. The heat energy *Q* cannot be converted back completely into the kinetic energy $E_{k}$ of the car and a useful work cannot be obtained from thermal energy without a temperature difference $T_{he}-T_{co}$ according to this law. Therefore, we have to burn fuel in the heat engines to create the temperature difference $T_{he}-T_{co}$.

We would not have to burn fuels if the law of entropy increase could be violated. Therefore Elliott Lieb and Jakob Yngvason [17] wrote that “*The world’s energy problems would be solved at one stroke*” at any reproducible violation of the second law of thermodynamics. The heat energy *Q* could be convert back in the kinetic energy $E_{k}$ of the car if all the laws of physics were reversible. However, a perpetuum mobile would be inevitable according to the law of energy conservation without the irreversibility postulated by the law of entropy increase.

Therefore, most scientists believe in this law. Arthur Eddington wrote: “*The second law of thermodynamics holds, I think, the supreme position among the laws of Nature. If someone points out to you that your pet theory of the universe is in disagreement with Maxwell’s equations—then so much the worse for Maxwell’s equations. If it is found to be contradicted by observation, well, these experimenters do bungle things sometimes. But if your theory is found to be against the second law of thermodynamics I can give you no hope; there is nothing for it but collapse in deepest humiliation*” [18].

### 2.2. The Centuries-Old Belief in the Impossibility of a Perpetuum Mobile

The supreme position of the second law of thermodynamics is definitely connected with the centuries-old belief of scientists in the impossibility of a perpetuum mobile. The perpetuum mobile is one of the oldest problems of science. The first attempts to invent a perpetuum mobile were known beginning with the 13th century, but already Leonardo da Vinci (1452–1519) stigmatized its inventors equally with alchemists. Stivin (1548–1620) postulated first the principle of impossibility of any perpetuum mobile. Because of such history, most scientists interpret perpetuum mobile as a problem of the ignorant Middle Ages on a level with alchemy.

The Paris Academy of Sciences decided in 1775 to not consider any project of perpetuum mobile. Some scientists still refer to this decision when it comes to a violation of the second law of thermodynamics. Such references can hardly have a scientific basis, since the Parisian academicians could not have known about, in 1775, not only, for example, quantum mechanics, but even thermodynamics. Nevertheless, the centuries-old belief in the impossibility of a perpetuum mobile has played an important role in the dramatic history of the second law of thermodynamics. The belief in the impossibility of a perpetuum mobile has persisted despite several fundamental changes in our ideas about nature over the past centuries. One of the most fundamental changes in the ideas of most scientists about thermodynamics processes occurred in the beginning of the 20th century.

## 3. The Struggle between Thermodynamic-Energy and Atomistic–Kinetic Worldviews

Smoluchowski wrote in his article “Limits of Validity of the Second Law of Thermodynamics” published in 1914: “*I begin my presentation of the above topic with a brief historical overview. Anyone who has been involved in the struggle between thermodynamic-energy and atomistic-kinetic worldviews for the past forty years knows why I do this. It is no longer easy for us to imagine the way of thinking that prevailed at the end of the last century. After all, at that time, scientists in Germany and France were convinced that the kinetic theory of atoms had already played a role. The principle of Carnot, intuitively understood by him, we call since the time of Clausius the second law of thermodynamics. Because of the confidence in the great achievements of thermodynamics, this principle has been elevated to the rank of the absolute, exact dogma without exclusion. And since at that time molecular kinetics in the interpretation of this principle faced certain difficulties associated with the irreversibility of processes, it, together with atomistics, was immediately condemned as untenable. Although Boltzmann tried to prove that if there are contradictions, they still cannot practically become tangible*” [16].

Smoluchowski made a very important remark that the way of thinking in particular about the second law of thermodynamics in 1914 differed fundamentally from the one that prevailed at the end of the 19th century. Most scientists in the late 19th and even early 20th century negatively related to the Maxwell–Boltzmann statistical theory because of its contradiction with the second law of thermodynamics. Many scientists, supporters of the thermodynamic-energy worldview, for example Wilhelm Ostwald, Nobel Prize Winner 1909, denied even the existence of atoms and their perpetual thermal motion. Smoluchowski pointed out one of the main objections to the atomistic-kinetic theory: the laws of mechanics describing the thermal motion of atoms are reversible, while the second law of thermodynamics postulates irreversibility as a necessary condition for the impossibility of a perpetuum mobile.

### 3.1. The Difference in the Understanding of Irreversibility between Scientists of the 19th and 20th Centuries

Scientists of both the 19th and 20th centuries understood that the law of entropy increase postulates irreversibility in nature. However, the understanding of irreversibility by the most scientists of the 19th century differed fundamentally from the one in the 20th century. Smoluchowski wrote in 1914 about how irreversibility was understood in the 19th century: “*In the phenomena of fluctuation experimentally observed in recent years, it seems extremely strange to a supporter of classical thermodynamics that he sees with his own eyes the reverse course of processes that are generally regarded as irreversible. Because according to the classical theory, the second principle of thermodynamics should disappear if at least one process, regarded as irreversible, admits reversibility*” [16].

Supporters of classical thermodynamics denied the possibility of any motion at thermodynamic equilibrium, in particular the perpetual thermal motion of atoms. The investigations of such fluctuation phenomena as the Brownian motion by Einstein, Smoluchowski and others have convinced even supporters of the thermodynamic-energy worldview in the existence of perpetual thermal motion of atoms, molecules, electrons, ions, and other particles. None of the modern scientists doubt that the random motion of particles suspended in a medium, a liquid or a gas, first described by the botanist Robert Brown as far back as in 1827, is a result of the perpetual thermal motion of atoms or molecules of the liquid or the gas.

Among the great scientists, Albert Einstein was one of the first who modeled, in 1905, the motion of the pollen particles as a result of the thermal motion of individual water molecules [19]. This explanation of Brownian motion was further verified experimentally by Jean Perrin and other experimenters. Perrin was awarded the Nobel Prize in Physics in 1926 “*for his work on the discontinuous structure of matter*”. The Brownian motion is the result of the many-body interactions. No model accounting for every involved molecule can describe this motion. Therefore Einstein, Smoluchowski and others used the statistical theory proposed in the 19th century by Maxwell, Boltzmann, Gibbs, and others for the description of the Brownian motion. The success of this description was one of the main reasons for the victory of the atomistic-kinetic worldview over the thermodynamic-energy worldview in the 20th century.

### 3.2. The Law of Chaos Increase

The interpretation of the law of entropy increase changed fundamentally after this victory: this law began to be understood as the law of chaos increase. Modern scientists follow Maxwell, Boltzmann and Gibbs in this interpretation. J.C. Maxwell wrote in 1878 [20] that “*the second law is drawn from our experience of bodies consisting of an immense number of molecules*”. And Joel Lebowitz wrote approximately the same after more than century: “*It is not every microscopic state of a macroscopic system that will evolve in accordance with the second law, but only the ’majority’ of cases—a majority which however becomes so overwhelming when the number of atoms in the system becomes very large that irreversible behavior becomes a near certainty*” [21].

Smoluchowski wrote in 1914: “*Thus, the issue is considered very different today than it was twenty years ago. Atomistics is recognized as the basis of modern physics in general; the second law of thermodynamics has once and for all lost its significance as an unshakable dogma, as one of the basic principles of physics*” [16]. However, the dogma has changed rather than lost its significance. The victory of the atomistic–kinetic worldview has created the illusion that the law of entropy increase and thus, the irreversibility in nature belong to our a priori rather than empirical knowledge. The gas of molecules or atoms are distributed evenly in a closed box after the partition separating the two parts of the box with different gas pressure is removed. It is difficult to doubt in this case that the Boltzmann entropy
(2)$$S=-k_{B}\sum_{i}p_{i}\ln p_{i}\;$$
has increased and its decrease to the initial value is extremely unlikely. Here, $p_{i}$ is the probability that the system is in a *i* state; $k_{B}$ is the Boltzmann constant.

The law of chaos increase is based on our a priori rather than empirical knowledge. Thus, the kinetic theory of atoms, rejected by many physicists in the 19th century because of the contradiction with the irreversibility postulated by the second law of thermodynamics, gave an a priori argument to justify the belief in the impossibility of a perpetuum mobile. Most scientists do not doubt in this argument in spite of the reversibility paradox, formulated by Johann Loschmidt and William Thomson, and the recurrence paradox, formulated by Henri Poincare and Ernst Zermelo.

## 4. The Assumption of Molecular Disorder Belong to Our Empirical Rather Than a Priori Knowledge

The reversibility paradox, i.e., the contradiction between the reversibility of mechanics and the irreversibility of thermodynamics, was considered as important objections against Boltzmann’s H-theorem, which is a mathematical substantiation of the law of chaos increase. However, most scientists are sure that the Boltzmann H-theorem overcame this contradiction. C.G. Weaver argues in a recent publication [22] that the reversibility paradox can be resolved with the help of the hypothesis of molecular chaos. However, he does not discuss the validity of the hypothesis itself.

This hypothesis is crucial to the modern belief in the law of entropy increase. Hendrik Lorentz recognized in 1887 [23] that this hypothesis is necessary for the proof of Boltzmann’s H-theorem. Max Planck questioned the H-theorem because of the groundlessness of this hypothesis, or rather the assumption. He noted in his Scientific Autobiography that “*Boltzmann omitted in his deduction every mention of the indispensable presupposition of the validity of his theorem namely, the assumption of molecular disorder. He must have simply taken it for granted*” [24]. Rather, most scientists of the 20th century other than Boltzmann have taken the assumption of molecular disorder for granted. Boltzmann understood the need for the assumption of molecular disorder. He wrote in 1896: “*We shall now explicitly make the assumption that the motion is molar- and molecular-disordered, and also remains so during all subsequent time*” [25].

However, Boltzmann did not substantiate the universality of the assumption of molecular disorder in any way. Therefore, Planck was right to doubt this assumption. The assumption of molecular disorder needed for the modern substantiation of the law of entropy increase implies that no ordered thermal motion of atoms, molecules, electrons, ions, Brownian particles and other particles is possible. We cannot be sure a priori that such a motion is impossible. Therefore the assumption of molecular disorder cannot belong to our a priori knowledge. Empirically, the assumption of the impossibility of ordered thermal motion can only be refuted, but not proven.

### 4.1. The Assumption of Molecular Disorder Is Needed for the Impossibility of an Useful Perpetuum Mobile

Noting that, with the victory of the atomistic–kinetic worldview, the second law of thermodynamics lost its importance as an unwavering dogma, Smolukhovsky argued: “*On the contrary, from the point of view of molecular statistics, the position of thermodynamics about the impossibility of perpetuum mobile of the second kind is correct, if this expression is given a more precise meaning, namely: an automatic machine continuously making work by consuming the heat of another body with a lower temperature*” [16].

Smoluchowski stated that “*the widespread opinion that molecular fluctuations could be used directly to construct a simple perpetuum mobile is completely wrong*” [16] and was proving his statement with the help of the following example: “*First, let’s imagine a gummigut particle, which, despite its gravity, will remain suspended above the bottom of the vessel due to Brownian motion. We will attach to the bottom of the vessel a vertical rail with sawtooth teeth or, better, a rail in which a zigzag groove is cut; at a certain height in each groove we will adapt a sleeve that is pressed by an elastic spring on one side, so that due to the latching of this device, the particle in the normal state can move only from the bottom up, but not in the opposite direction*” [16].

Smoluchowski understood that the impossibility of directed thermal motion is needed for the impossibility of an useful perpetuum mobily. However, he, in contrast to the great scientist Max Planck, did not understand that the assumption of molecular disorder cannot have a scientific substantiation. Smoluchowski proved that a useful perpetuum mobile cannot be created on the base of the mechanical machine considered in his example since all parts of this mechanical machine move chaotically due to thermal fluctuations.

The famous physicist Richard Feynman understood also that the Carnot principle can be valid only if no directed thermal motion is possible. He repeated in Chapter 46 “Ratchet and pawl” of his lectures on physics [26] the Smoluchowski proof that no useful work can be obtained from heat energy without a temperature difference $T_{he}-T_{co}$ with the help of a mechanical machine. This ’proof’ rather testifies to the belief of Smoluchowski, Feynman and most physicists in the impossibility of a perpetuum mobile than a real universal proof of the assumption of molecular disorder.

### 4.2. The Chaotic of the Nyquist Current as a Consequence of the Assumption of Molecular Disorder

The centuries-old belief in the impossibility of a perpetuum mobile turned out to be so persistent that even paradoxical quantum phenomena could not question the assumption of molecular disorder. Many scientists are ready to abandon realism, even macroscopic [27,28,29], because of the paradoxical nature of some quantum phenomena. However, only a few scientists [30] are willing to question the law of entropy increase because of these phenomena. Although the latter has more grounds already due to quantization, which is not only observed in atoms.

A useful work cannot be obtained from the Brownian motion because of its disorder. According to the equation
(3)$$m\frac{dv}{dt}=F_{\mathrm{dis}}+\chi \left(t\right)\;$$
proposed by Paul Langevin in 1908 [31], a Brownian particle, with a small mass *m*, moves under the influence of a random force $\chi =(\chi_{x},\chi_{y},\chi_{z})$, representing the effect of the collisions with the molecules of the fluid, and is decelerate by a viscous force (friction force or dissipation force) $F_{\mathrm{dis}}=-\lambda v$ proportional to the velocity of the particle $v=(v_{x},v_{y},v_{z})$.

The random force Ø has a Gaussian probability distribution with correlation function
(4)$$<\chi_{i}\left(t\right)\chi_{j}\left(t^{\prime }\right)>=2\lambda k_{B}T\delta_{i,j}\delta \left(t-t^{\prime }\right)\;$$

The delta-function form of the correlations $\delta_{i,j}$, $\delta (t-t^{\prime })$ in (4) is the mathematical expression for the assumption of molecular disorder: the i-th component of the vector Ø are completely uncorrelated between themselves, and the Langevin force at a time *t* is completely uncorrelated with the force at any other time $t^{\prime }$. The dependence of the correlation function of the random Langevin force (4) from the damping coefficient λ is known as the Einstein relation.

The Langevin Equation (Equation 3) describes enough well the observed motion of Brownian particles, which is random. The average velocity and the average Langevin force of such motion equal zero. However, can we be sure a priori that the average velocity of any type of Brownian motion should always be zero? We can be a priori sure that the average velocity can be nonzero only in a circular motion, since Brownian motion is observed at thermodynamic equilibrium, in which no macroscopic transfer of mass or energy from one region of space to another can be. The Nyquist [32] (or Johnson [33]) noise current
(5)$$\bar{I_{Nyq}^{2}}=\frac{k_{B}T\Delta \omega }{R}\;$$
observed in a metal ring with a non-zero resistance R>0 is the example of the circular Brownian motion [26].

The Nyquist noise cannot be used to obtain useful work at thermodynamic equilibrium, since all elements of any electrical circuit have the same distribution of the power $P_{Nyq}=k_{B}T\Delta \omega$ in frequency ω. This distribution is a consequence of the randomness of this type of Brownian motion. The power $P_{Nyq}=0$ at the zero frequency ω=0 and the average value of the Nyquist current $\bar{I_{Nyq}}=0$ equal zero also because of the randomness.

### 4.3. Quantization and the Persistent Current

The absence of the directed component in the average value of the Nyquist current $\bar{I_{Nyq}}=0$ can be explained a priori by the equal probability of the current with the opposite direction, $\bar{I_{Nyq}}=\int_{0}^{Imax}dI\left(P\left(+I\right)I-P\left(-I\right)I\right)=0$ at P(+I)=P(−I). However, this explanation is valid only according to classical physics when all states in the ring are permitted. According to quantum mechanics, some states of a particle with a mass *m* and a charge *q* can be forbidden in the ring with a radius *r* because of the quantization of the canonical momentum p=mv+qA
(6)$$\oint_{l}dlp=\oint_{l}dl\left(mv+qA\right)=n2\pi \hbar \;$$

The velocity of such a particle
(7)$$\oint_{l}dlv=\frac{2\pi \hbar }{m}\left(n-\frac{\Phi }{\Phi_{0}}\right)\;$$
cannot be zero when the magnetic flux inside the ring $\Phi =\oint_{l}dlA$ is not divisible $\Phi \neq n\Phi_{0}$ by the flux quantum $\Phi_{0}=2\pi \hbar /q$ and the permitted velocity $v=\left(\hbar /mr\right)\left(n-\Phi /\Phi_{0}\right)$ in a one direction corresponds the forbidden velocity in the opposite direction at $\Phi \neq n\Phi_{0}$ and $\Phi \neq \left(n+0.5\right)\Phi_{0}$. This asymmetry is the cause of such well known quantum phenomenon as the persistent current observed in superconductor [34,35,36] and normal metal [37,38] rings. The persistent current $I_{p}$ is observed at the thermodynamic equilibrium, like the Nyquist noise current (5), and does not decay in spite of a non-zero resistance of both superconductor ring [35,36] and normal metal ring [37,38]. However, the directed component of the persistent current $\bar{I_{p}}$, i.e., the current at the zero frequency ω=0, is not zero $\bar{I_{p}}=I_{p}\left(\omega =0\right)\neq 0$ [34,35,36,37,38] in contrast to the Nyquist noise current (5).

Obviously, this fundamental difference the persistent current from the Nyquist current (5) became possible due to the violation of symmetry between the opposite directions due to quantization of the canonical momentum (6). This is confirmed by observations: $\bar{I_{p}}\neq 0$ at $\Phi \neq n\Phi_{0}$ and $\Phi \neq \left(n+0.5\right)\Phi_{0}$ [35,37,38] when there is the asymmetry between the opposite directions (7). The quantization (7) can be visible unless the energy difference between the permitted states is much less than the energy of thermal fluctuations $k_{B}T$. The energy difference between the permitted states of a quantum particle
(8)$$\Delta E_{n+1,n}=\frac{mv_{n+1}^{2}}{2}-\frac{mv_{n}^{2}}{2}\approx \left(2n+1\right)\frac{\hbar^{2}}{2mr^{2}}\;$$
decreases with the increases of the radius of the quantization *r*. The energy difference $\Delta E_{n+1,n}\approx 5\times 10^{-27}$ J of single electron with the mass $m=9\times 10^{-31}$ kg in a ring with the real radius $r=10^{-6}$ m = 1 μm corresponds to very low temperature $T<\Delta E_{n+1,n}/k_{B}\approx 0.0004$ K. Therefore, the persistent current of electrons in normal metal rings was reliably observed relatively recently [37,38], almost forty years after the prediction [39] of this paradoxical quantum phenomenon.

The persistent current Cooper pairs in a superconductor with a non-zero resistance R>0 was observed first [6] almost fifty years before the one in normal metal since the spectrum of superconducting condensate is much more discrete than the one of electrons [40]. Cooper pairs, being bosons unlike electrons, are at the same level *n* and have the same velocity (7). Therefore, the persistent current of Cooper pairs
(9)$$I_{p}=qsn_{s}v_{n}=q\frac{N_{s}}{2\pi r}v_{n}=\frac{\Phi_{0}}{L_{k}}\left(n-\frac{\Phi }{\Phi_{0}}\right)\;$$
is much greater than the persistent current of electrons and the discreteness of the energy spectrum
(10)$$E_{k}=N_{s}\frac{mv_{n}^{2}}{2}=\frac{L_{k}I_{p}^{2}}{2}=I_{p,A}\Phi_{0}{\left(n-\frac{\Phi }{\Phi_{0}}\right)}^{2}\;$$
is greater by a factor equal to the number of Cooper pairs $N_{s}=2\pi rsn_{s}$ [40] in the ring with the pair density $n_{s}$, the radius *r* and the small cross-section $s\ll \lambda_{L}^{2}$. $L_{k}=m2\pi r/sq^{2}n_{s}=\left(\lambda_{L}^{2}/s\right)\mu_{0}2\pi r$ is the kinetic inductance of the ring; $\lambda_{L}={\left(m/\mu_{0}q^{2}n_{s}\right)}^{0.5}$ is the quantity generally referred to as the London penetration depth. Its typical value equals $\lambda_{L}$ = 50 nm = $5\times 10^{-8}$ m.

The persistent current oscillates in magnetic field *B* with the period $B_{0}=\Phi_{0}/\pi r^{2}$ corresponding to the magnetic flux $\Phi_{0}=B_{0}\pi r^{2}$ inside the ring and the amplitude $I_{p,A}=\Phi_{0}/2L_{k}$ [34] when the quantum number *n* takes the value corresponding to the minimum of the kinetic energy (10). The quantum oscillations are observed [34,35,36] since the energy (10) difference $|E_{k}\left(n+1\right)-E_{k}\left(n\right)|\approx I_{p,A}\Phi_{0}$ is very large. The energy difference at $\Phi =n\Phi_{0}$ corresponds the temperature $I_{p,A}\Phi_{0}/k_{B}\approx 2000$ K [41] at a typical value of the amplitude of the persistent current $I_{p,A}\approx 10\;\mu$A observed in a superconducting ring even with the small cross-section $s\ll \lambda_{L}^{2}$ [34]. The discreteness increases with the increase of all three sizes of the ring $|E_{k}\left(n+1\right)-E_{k}\left(n\right)|\approx n_{s}s2\pi r\left(\hbar^{2}/2mr^{2}\right)\propto \left(s/r\right)$ since the energy difference is proportional to the number of Cooper pairs and, thus, the volume of the ring $|E_{k}\left(n+1\right)-E_{k}\left(n\right)|\propto N_{s}=2\pi rsn_{s}\propto 2\pi rs$ [40]. That is why superconductivity is a macroscopic quantum phenomenon.

### 4.4. The Average Value of the Langevin Force Can Be Nonzero Due to the Quantization

The quantization (6) of angular momentum pr=nℏ was postulated in 1913 by Bohr for the explanation of the discrete spectrum of electrons in atom. The discreteness in atom is much larger than in a real ring because of the smallness of the radius of the quantization. Normal metal ring is fundamentally different from an atom, since the electrons in the metal are scattered by defects and phonons, unlike atomic orbits, and therefore the ring has a non-zero resistance R>0. The value of the resistance *R* determines the value of a relaxation time $\tau_{RL}=L/R$ during which any electric current must decay in the ring with the total inductance $L=L_{f}+L_{k}$. The electric current should not decay only in the case of the perfect conductivity, when the resistance R=0. In accordance with the electrodynamics law $L_{k}dI_{p}/dt=-d\Phi /dt$, the increase of magnetic flux inside a ring with R=0 from Φ=0 to $\Phi =BS-L_{f}I_{p}$ should induce the electric current $I_{p}=-\Phi /L_{k}$, which cannot decay arbitrarily long time, until R=0. The electric current does not decay for an arbitrarily long time in a superconducting ring.

The experimental results [6,35,36,37,38] testify that the persistent current can be observed for an arbitrarily long time not only at R=0, but also at R>0. In order to understand the cause of this paradoxical phenomenon, one should recall that the electric current $I_{p}=-\Phi /L_{k}$ can appear in a superconducting ring in two ways: (1) first the temperature is lowered below the temperature of superconducting transition $T<T_{c}$ in zero magnetic field H=0 and thereafter the external magnetic field is increased up to $H=B/\mu_{0}$; (2) first the magnetic field is increase up to $H=B/\mu_{0}$ at $T>T_{c}$ and thereafter the temperature is lower down to $T<T_{c}$.

The current $I_{p}=-\Phi /L_{k}$ appears on the first way under the influence of the Faraday electromotive force −dΦ/dt≈SdB/dt because of the increase in the magnetic field in time. However, the current induced this force at $T>T_{c}$ will decay during a short relaxation time $\tau_{RL}=L/R$ on the second way because of a non-zero resistance R>0 in the normal state. According to the electrodynamics laws lowering the temperature from $T>T_{c}$ to $T<T_{c}$ in a magnetic field $H=B/\mu_{0}$ constant in time dB/dt=0 should not induce a current in the ring. Nevertheless, all measurements give experimental evidence that the electric current appears not only on the first way but also on the second way.

The electric current appears on the second way because of the quantization of the the canonical momentum p=mv+qA (6) since the velocity of Cooper pairs (7) cannot be equal zero at $\Phi \neq n\Phi_{0}$. This current is persistent since a superconducting state with zero current is forbidden at $\Phi \neq n\Phi_{0}$. The quantization replaces the Faraday electromotive force on the second way and as consequence the same current $I_{p}=-\Phi /L_{k}$ appears on the both ways at $\left|\Phi \right|\approx |BS|<\Phi_{0}/2$ when the quantum number corresponding to the minimum energy equals zero n=0. We consider here rings with the small cross-section $s\ll \lambda_{L}^{2}$ used in the experimental works [34,35,36]. The magnetic inductance $L_{f}$ of such rings is smaller than the kinetic inductance $L_{f}\approx \mu_{0}2\pi r\ll L_{k}=\left(\lambda_{L}^{2}/s\right)\mu_{0}2\pi r$ and the magnetic flux $L_{f}I_{p}$ induced by the persistent current $I_{p}$ is small $L_{f}I_{p}\ll \Phi_{0}$.

The velocity of the mobile charge carriers changes on the same value on the both ways at $\left|\Phi \right|\approx |BS|<\Phi_{0}/2$, whereas the canonical momentum p=mv+qA changes only on the second way since the velocity on the first way increases in accordance the Newton second law mdv/dt=qE=−qdA/dt according to which dp/dt=mdv/dt+qdA/dt=0. The canonical momentum should change on the second way on the value $\Delta p=\left(\hbar /r\right)\left(n-\Phi /\Phi_{0}\right)$ since the angular momentum equals $rp=rqA=\hbar \Phi /\Phi_{0}$ in the normal state at $H=B/\mu_{0}=\Phi /\mu_{0}S\neq 0$ because of the zero velocity v=0 and should have the quantum value pr=nℏ in the superconducting state. This change of the canonical momentum and the velocity because of the quantization (6) should occur at each comeback of the ring in the superconducting state when $\Phi \neq n\Phi_{0}$. This change because of the quantization during a time unity at the switching of the ring between normal and superconducting states with a frequency $f_{sw}=N_{sw}/\Theta$
(11)$$F_{q}=\hbar \frac{\left(\bar{n}-\Phi /\Phi_{0}\right)}{r}f_{sw}\;$$
was called quantum force in [42]. $\bar{n}=\int_{\Theta }dtn/\Theta =\sum_{i=1}^{i=N_{sw}}n_{i}/N_{sw}$ is the average value of the quantum number after $N_{sw}\gg 1$ comeback of the ring into the superconducting state during a long time $\Theta \gg 1/f_{sw}$.

The ring can be switched between superconducting and normal states because of different reasons: temperature change, non-equilibrium noise, external AC current and other external factors. However, only thermal fluctuations can switch the ring at thermodynamic equilibrium in a narrow temperature region $\Delta T_{fl}$ near the superconducting transition $T\approx T_{c}$ [43,44]. The persistent current of Cooper pairs is observed at a non-zero resistance R>0 [6,35,36] in this and only in this fluctuation region: at $T>T_{c}+\Delta T_{fl}$, above this region $R=R_{n}>0$ but $I_{p}=0$ whereas at $T<T_{c}-\Delta T_{fl}$, below this region $I_{p}\neq 0$ but R=0. This experimental fact reveals that this paradoxical phenomenon is a fluctuation phenomena, i.e., a type of the circular Brownian motion, like the Nyquist noise current in a metal ring (5).

The action of the Langevin force in this Brownian motion is manifested in the increase in the velocity of $N_{s}$ mobile charge carriers up to the permitted value (7) after switching the ring by fluctuations to the superconducting state with the number of Cooper pairs $N_{s}$. The average velocity of these mobile charge carriers decreases down to zero under the action of the dissipation force after their switching by fluctuations to the normal state. The angular momentum of each mobile charge carrier changes on $\hbar (n-\Phi /\Phi_{0})$ at each switching in the superconducting state. Therefore the average value of the Langevin force should be equal $\bar{\chi }\approx \bar{N_{s}}F_{q}$, where $\bar{N_{s}}=2\pi rs\bar{n_{s}}$ is the average number of the Cooper pairs in the ring which decreases rapidly with the temperature increase at $T\approx T_{c}$ [43,44], but remains a large number even at $T>T_{c}$, where the persistent current was predicted [45] and observed [46].

The spectrum of the permitted states remains discrete even in the fluctuation region where R>0. The energy (10) difference $|E_{k}\left(n+1\right)-E_{k}\left(n\right)|\approx I_{p,A}\Phi_{0}$ at the amplitude $I_{p,A}\approx 0.1\;\mu A$ of the oscillations of the persistent current observed [35,36] in the fluctuation region at $T\approx T_{c}$ corresponds to the temperature $I_{p,A}\Phi_{0}/k_{B}\approx 20\;K$. Thus, the persistent current is observed at a non-zero resistance $0<R<R_{n}$ near superconducting transition [6,35,36] since it is the directed Brownian motion [42]. The average value of the Langevin force $\bar{\chi }\approx \bar{N_{s}}F_{q}$ is not zero because of the discreteness of the permitted state spectrum even in the fluctuation region.

### 4.5. The Persistent Voltage

The persistent current observed at R>0 can be used, in contrast to the Nyquist noise (5), for an useful work since its power $RI_{p}$ is not zero at the zero frequency ω=0. It is well known that the potential difference
(12)$$V=0.5(R_{n}-R_{w})I\;$$
is observed on the halves of the ring with different resistance $R_{B}>R_{A}$ when the conventional circular electric current *I* is induced by the Faraday electric field $RI=(R_{n}+R_{w})I=-d\Phi /dt$. It was shown firstly in [47] that quantum mechanics predicts a possibility to observe the potential difference with a directed component $V_{dc}$ proportional to the average value of the persistent current $\bar{I_{p}}\left(\Phi \right)=2I_{p,A}\left(\bar{n}-\Phi /\Phi_{0}\right)$ when an asymmetric ring or its segments are switched between superconducting and normal states, see also [12]. Such magnetic oscillations of the DC voltage $V_{dc}\left(\Phi \right)\propto \bar{I_{p}}\left(\Phi \right)=2I_{p,A}\left(\bar{n}-\Phi /\Phi_{0}\right)$ were observed firstly thirty years before this prediction at the measurements of an asymmetric dc SQUID, i.e., a superconducting loop with two Josephson junctions [48]

A.Th.A.M. De Waele et al. [48] have not understood the importance of the paradoxical effect they discovered because of the belief in the law of entropy increase. They were concerned only with the question about a source of the power $V_{dc}\bar{I_{p}}=V_{dc}^{2}/R$ that they observed at $\Phi \neq n\Phi_{0}$ and $\Phi \neq \left(n+0.5\right)\Phi_{0}$. They assumed that the source of this power is the emitted electromagnetic radiation of broadcasting stations and have confirmed that the visible dc voltage $V_{dc}\left(\Phi \right)\propto \bar{I_{p}}\left(\Phi \right)$ disappears when all parts of the equipment are shielded more carefully [48].

Measurements carried out on asymmetric aluminum rings confirmed that below the temperature of the superconducting transition $T<T_{c}$, quantum oscillations of the dc voltage $V_{dc}\left(\Phi \right)\propto \bar{I_{p}}\left(\Phi \right)$ are induced by a non-equilibrium noise [49] and can be induced by an alternating current [50]. These measurements revealed that the dependence of the amplitude $V_{A}$ of the DC voltage oscillations in magnetic field $V_{dc}\left(\Phi \right)\propto \bar{I_{p}}\left(\Phi \right)$ on the amplitude $I_{0}$ of the AC current $I_{ac}=I_{0}\sin\omega t$ or noise is non-monotonic: the DC voltage $V_{dc}\left(\Phi \right)$ appears when the amplitude $I_{0}$ reaches the value of the critical current of the ring $I_{0}\approx I_{c}\left(T\right)\approx I_{c}\left(0\right){\left(1-T/T_{c}\right)}^{3/2}$ at the measurement temperature *T*, the amplitude $V_{A}$ increases rapidly to a maximum with the increase of $I_{0}$, and decreases with further increase of $I_{0}$.

Reducing the critical current $I_{c}\left(T\right)\approx I_{c}\left(0\right){\left(1-T/T_{c}\right)}^{3/2}$ to zero as the temperature approaches the critical temperature $T\to T_{c}$ means that arbitrarily weak noise can induce the DC voltage $V_{dc}\left(\Phi \right)$. However, the amplitude of the oscillations of the persistent current $I_{p,A}\left(T\right)\approx I_{p,A}\left(0\right)\left(1-T/T_{c}\right)$ reduces also at $T\to T_{c}$. Therefore, the visible oscillations of the DC voltage $V_{dc}\left(\Phi \right)\propto \bar{I_{p}}\left(\Phi \right)$ are observed [36,51,52] only a narrow temperature region at a give value of the amplitude $I_{0}$ of noise or the AC current: below this region $I_{0}<I_{c}\left(T\right)\approx I_{c}\left(0\right){\left(1-T/T_{c}\right)}^{3/2}$, and therefore the noise or the AC current cannot switch the ring in the normal state whereas above this region the DC voltage $V_{dc}\left(\Phi \right)\propto \bar{I_{p}}\left(\Phi \right)$ is not visible because small value of the persistent current.

Measurements of single asymmetric ring [49] and system of such identical rings connected in series [36,52,53] have revealed that the reduction of the non-equilibrium noise with the help of careful shielding reduces rather than annihilates the DC voltage $V_{dc}\left(\Phi \right)\propto \bar{I_{p}}\left(\Phi \right)$. The amplitude $V_{A}$ of $V_{dc}\left(\Phi \right)$ increases proportional to the number of the identical asymmetric rings connected in series [50] since the DC power adds up, in contrast to the chaotic Nyquist voltage. For example, the oscillations $V_{dc}\left(\Phi \right)$ with the maximum amplitude $V_{A}\approx 1.5\;\mu$V was observed on a single asymmetric ring and with $V_{A}\approx 1500\;\mu$V on a system of 1080 such rings connected in series at the same amplitude $I_{0}\approx 3\;\mu$A of the AC current and approximately the same temperature [54]. Therefore, arbitrarily weak noise, up to equilibrium noise, can be detected using a system with a sufficiently large number of asymmetric superconducting rings [51].

The oscillations $V_{dc}\left(\Phi \right)$ with the maximum amplitude $V_{A,1080}\approx 0.1\;\mu$V = $10^{-7}$ V were observed on the system of 1080 asymmetric rings (the amplitude $V_{A,1}\approx 10^{-10}$ V on each ring) when the screening of non-equilibrium noise was particularly thorough [52]. The observation of $V_{dc}\left(\Phi \right)$ in the temperature region corresponding to the resistive transition where R>0 [52] allows us to assume that this DC voltage was induced by the switching of the rings between superconducting and normal state at thermodynamic equilibrium. However, it should be emphasized that the observations [36,48,49,51,52,53] of the DC power $V_{dc}\bar{I_{p}}=V_{dc}^{2}/R$ challenges to the law of entropy increase in any case.

A.Th.A.M. De Waele et al. [48] did not take into account that the DC voltage $V_{dc}\left(\Phi \right)\propto \bar{I_{p}}\left(\Phi \right)$ is observed in a magnetic field constant in time, i.e., without the Faraday electromotive force −dΦ/dt. Therefore, the persistent current flows against the total electric field $E_{t}=E_{p}=-\nabla V_{dc}$ in one of the ring halves [55], in contrast to the conventional circular current *I* (12), which flows against the potential electric field $E_{p}=-\nabla V$, but not against the total electric field $E_{t}=E_{p}+E_{F}=-\nabla V-dA/dt$. This paradox can be explain only if we take into account that the quantum force (11) replaces the Faraday electromotive force and counteract to the dissipation force [56] when rings are switched between superconducting and normal states.

The persistent current, having a macroscopic kinetic energy (10) $E_{k}\gg k_{B}T$ at $T<T_{c}$, is damped with the generation of Joule heat during a short relaxation time $\tau_{RL}=L/R$ after the switching of the ring into the normal state. The dissipation of the kinetic energy (10) $E_{k}\gg k_{B}T$ into Joule heat $Q=E_{k}$ is an irreversible thermodynamic process at which the entropy increases on a macroscopic value $\Delta S=Q/T=E_{k}/T\gg k_{B}$. Therefore, the appearance of the persistent current because of the quantization (6) at each return of the ring in superconducting state is a process reverse to the irreversible thermodynamic process during which Joule heat *Q* is converted back into the kinetic energy (10) of the persistent current (9) and the entropy decreases on the macroscopic value $\Delta S=-Q/T=-E_{k}/T$. The impossibility of such reverse process is the essence of the law of entropy increase.

According to the Carnot Equation (Equation 1), a temperature difference is needed in order to create the directional movement with the help of a heat engine. However, the persistent current $\bar{I_{p}}\left(\Phi \right)$ is already the directional movement observed [35,36,37,38] at thermodynamic equilibrium in the absence of a temperature difference. The persistent voltage $V_{dc}\left(\Phi \right)\propto \bar{I_{p}}\left(\Phi \right)$ observed on the halves of the asymmetric rings can be used to perform a useful work in the absence of a temperature difference due to a non-zero value of its power $P_{dc}=V_{dc}\bar{I_{p}}=V_{dc}^{2}/R$ at the zero frequency ω=0. The Nyquist noise cannot be used for a useful work without a temperature difference since each element of the electrical circuit has the same power $P_{Nyq}=k_{B}T\Delta \omega$ in any frequency range Δω.

Therefore, the power $P_{Nyq}=k_{B}T\Delta \omega$ will always flow from a hot body to a cold body, in accordance with the law of entropy increase, no matter what frequency filters are used. The DC power can be transferred from a cold body to a hot body when the filters pass only the DC current. The DC voltage $V_{dc}\left(\Phi \right)\propto \bar{I_{p}}\left(\Phi \right)$ induced on 1080 asymmetric rings at the temperature T≈1 K is observed at the room temperature T≈300 K in the experimental work [54]. The DC power $P_{dc}=V_{dc}\bar{I_{p}}=V_{dc}^{2}/R$ is observed in [54] since the directed electric current $\bar{I_{p}}$ flows against the directed electric field $E=-\nabla V_{dc}$ in one of the halves of asymmetric superconducting ring due to the violation of the assumption of molecular disorder in the quantum phenomenon of the persistent current.

### 4.6. The Persistent Current Observed at a Nonzero Resistance Refutes Experimentally the Assumption of Molecular Disorder

The observation of the persistent current in rings with nonzero resistance at thermodynamic equilibrium [6,35,36,37,38] is an obvious experimental refutation of the assumption of molecular disorder, the validity of which was questioned by Planck [24]. This quantum phenomenon was first interpreted as a directed Brownian motion in the paper [42]. The directed Brownian motion is observed, contrary to the proof of its impossibility by Smolukhovsky [16] and Feynman [26], since the discreteness of the spectrum is not parts of a mechanical device that are subject to chaotic thermal motion [57].

The electric current *I* decays during a short relaxation time $\tau_{RL}=L/R$ at a non-zero resistance R>0 since its kinetic energy dissipates into Joule heat with the power $RI^{2}$. The electric current *I* can be observed at R>0 during a long time $t\gg \tau_{RL}$ if a power source $RI^{2}$ compensates the energy dissipation with the power $RI^{2}$. Therefore, the observations [6,35,36,37,38] of the persistent current $I_{p}\neq 0$ at R>0 during a long time $t\gg \tau_{RL}$ proves the existence of a DC power source $RI^{2}$ at thermodynamic equilibrium. The experimental evidence [6,35,36,37,38] of the existence of the DC power source $RI^{2}$ at thermodynamic equilibrium refutes the second law of thermodynamics.

There is only one way to save the faith in this law, to claim that the persistent current is a special electric current that can flow in rings with non-zero resistance R>0 without energy dissipation. This is exactly what A.C. Bleszynski-Jayich et al. [37] have made. They claim that the persistent current, they observed in normal metal rings, is dissipationless and write: “*A dissipationless equilibrium current flowing through a resistive circuit is counterintuitive, but it has a familiar analog in atomic physics: Some atomic species’ electronic ground states possess nonzero orbital angular momentum, which is equivalent to a current circulating around the atom*” [37].

The analogy with the atom is obviously false. Firstly, it is hardly possible to think that an electron creates a current in an atomic orbit, since an atom is centrally symmetric, unlike a ring. Secondly, there are no defects and phonons in the atom on which electrons are scattered in the metal ring. The electron’s elastic scattering length ≈40 nm, found by A.C. Bleszynski-Jayich et al. [37] in their normal metal rings, is less than the circumference of the rings 2πr≈2000÷5000 nm. A.C. Bleszynski-Jayich et al. [37] admit that the persistent current is produced by electrons diffusing around the ring. However, how can electrons diffuse in atoms? The analogy and the claim of A.C. Bleszynski-Jayich et al. [37] also contradict the experimental fact that the amplitude of the persistent current oscillations, measured by them, decreases exponentially with increasing temperature. Electronic orbits in atoms do not depend on temperature, and dissipationless current cannot depend on temperature since the absence of energy dissipation means the absence of energy exchange with the environment.

N.O. Birge agrees with A.C. Bleszynski-Jayich et al. [37], but recognizes: “*The idea that a normal, non-superconducting metal ring can sustain a persistent current - one that flows forever without dissipating energy - seems preposterous. Metal wires have an electrical resistance, and currents passing through resistors dissipate energy*” [58]. This idea is not only preposterous, but also contradicts elementary mathematics. A.C. Bleszynski-Jayich et al. [37] observe $I_{p}\neq 0$ and measure R>0, but they claim that $RI_{p}^{2}=0$. If A.C. Bleszynski-Jayich et al. and N.O. Birge [37,58] have paid attention to the reason for Planck’s doubts about the Boltzmann H-theorem, then perhaps they would not make the preposterous claim contradicting the elementary mathematics and experimental results. Only blind faith in the law of entropy increase could force A.C. Bleszynski-Jayich et al. and N.O. Birge [37,58] to make this preposterous claim. This faith has led physicists to much earlier make obvious mistakes and even to create false theories.

## 5. The Meissner Effect

According to almost universal belief from Maxwell’s time to the present day [21] “*the second law is drawn from our experience of bodies consisting of an immense number of molecules*” [20]. Quantization of electron orbits in an atom cannot challenge the second law of thermodynamics since thermodynamics does not describe individual atoms. The rings measured in the works [34,35,36,37,38] are bodies consisting of an immense number of atoms, electrons and/or Cooper pairs. Therefore the quantization (6) in the rings challenges the law of entropy increase. The discreteness of the spectrum of a superconducting ring (10) is proportional to the number $N_{s}=2\pi rsn_{s}$ of Cooper pairs in it. Therefore, quantization is observed in macroscopic superconducting structures, and superconductivity is a macroscopic quantum phenomenon.

### 5.1. Perfect Conductivity and Superconductivity

The first experimental evidence of this paradoxical fact is the Meissner effect discovered in 1933 [3]. Superconductivity was discovered in 1911 as perfect conductivity when Heike Kamerlingh Onnes observed for the first time that the resistance of some metals can abruptly decrease to zero at low temperatures. Physicists were sure before 1933 that the electric current can appear only in accordance with the Newton second law mdv/dt=qE and the electrodynamics laws. The current density $j=n_{s}qv$ in a perfect conductor with a density $n_{s}$ of the mobile carriers of a charge *q* should change in time $dj/dt=\left(n_{s}q^{2}/m\right)E=E/\mu_{0}\lambda_{L}^{2}$ under the action of an electric field *E*, according to these laws. According to the Maxwell equations rotE=−dB/dt, rotH=j and $B=\mu_{0}H$ the expression
(13)$$\lambda_{L}^{2}{▽}^{2}\frac{dH}{dt}=\frac{dH}{dt}\;$$
should be valid. According to (13), a change in the magnetic field *H* over time can penetrate in the perfect conductor only up to the penetration depth $\lambda_{L}$. The magnetic field inside a long cylinder with a macroscopic radius $R\gg \lambda_{L}$ equals
(14)$$h=H\exp-\frac{R-r}{\lambda_{L}}\;$$
after increasing the external magnetic field from 0 to *H* at $T<T_{c}$. The density of the surface screening current equals
(15)$$j=j_{0}\exp-\frac{R-r}{\lambda_{L}}\;$$
where $j_{0}=H/\lambda_{L}$ is the current density at r=R. The screening current (15) has the kinetic energy, the density of which $\varepsilon =n_{s}mv^{2}/2=\mu_{0}\lambda_{L}^{2}j^{2}/2$ corresponds to the energy
(16)$$E_{k}=\mu_{0}H^{2}\lambda_{L}\pi RL=\mu_{0}H^{2}V\frac{\lambda_{L}}{R}\;$$
of a macroscopic cylinder with the radius $R\gg \lambda_{L}$ and the length *L*.

Physicists understood that the surface screening current should dissipate in Joule heat within a short time after the transition in the normal state with a resistivity ρ>0, and were sure that this current cannot reappear when the cylinder will return to the superconducting state with ρ=0 in a magnetic field *H* constant in time dH/dt=0. Therefore, the transition of a bulk superconductor into the normal state in a magnetic field *H* was considered as an irreversible process at which “*the surface currents (15) associated with the field are damped with the generation of Joule heat*” [13].

However, W. Meissner and R. Ochsenfeld discovered in 1933 [3] that the surface screening currents (15) appear at the transition into the superconducting state in a magnetic field *H* constant in time dH/dt=0, contrary to the Newton second law and the electrodynamics laws. This experimental fact, because of which a superconductor differs from a perfect conductor, contradicts to the law of entropy increase if “*the surface currents associated with the field are damped with the generation of Joule heat*” [13] when the superconductivity is destroyed. The analogy of the screening current (15) with the movement of the car makes the contradiction more obvious.

The increase of the screening current (15) with an increase in the field $dj_{0}/dt=-\lambda_{L}^{-1}dH/dt$ under the influence of the Faraday electric field $E=-{\left(2\pi r\right)}^{-1}d\Phi /dt=-\mu_{0}\lambda_{L}dH/dt$ can be compared with the acceleration of a car by a driving force in the absence of a dissipation force. The car, like the current (15), can move indefinitely in the absence of the dissipation force and will quickly stop after the appearance of this force, like the current (15) after the transition into the normal state. The emergence of the persistent screening current (15) observed at the Meissner effect is analog to if a car began to move in the absence of a driving force, and uphill, since mobile charge carriers should accelerate against the action of the Faraday electric field $E=-{\left(2\pi r\right)}^{-1}d\Phi /dt$ in order to push the magnetic field *H* out of the superconductor, see (14).

If the car started moving uphill in the absence of a driving force, then everyone would consider it a miracle. However, scientists do not believe in miracles and therefore they do not consider the Meissner effect a miracle. Physicists in 1933 also did not believe that the law of entropy increase can be violated. Therefore, they concluded that the Meissner effect was able to turn the irreversible transition in which Joule heat is generated into a reversible transition in which no Joule heating can be. No one could explain how the persistent currents can be annihilated without the generation of Joule heat. Nevertheless, all experts were convinced after 1933 that the superconducting transition is a phase transition, i.e., a reversible thermodynamic process at which there no Joule heating can be. All theories of superconductivity were created on the basis of this confidence.

### 5.2. The Meissner Effect Is a Special Case of the Flux Quantization

From the very beginning, the Meissner effect was considered a quantum phenomenon, since this effect is impossible according to the laws of classical physics. Lev Landau was first who proposed an explanation of the Meissner effect as a quantum phenomenon. He proposed in 1941 [59] to describe superconducting state with the help of the wave function $\Psi_{GL}=\left|\Psi_{GL}\right|\exp i\varphi$ the square of the module of which $|\Psi_{GL}{|}^{2}=n_{s}$ describes the density of all superconducting particles whereas the phase gradient ℏ∇φ=p=mv+qA describes the canonical momentum of each particle. This wave function became the basis of the famous Ginzburg–Landau (GL) theory [1], which made it possible to describe many effects observed in superconductors as macroscopic quantum phenomena. The density of superconducting current equals
(17)$$j=\frac{q}{m}n_{s}\left(\hbar \nabla \varphi -qA\right)\;$$
according to [59] and the GL theory [1]. The GL Equation (Equation 17) allows deducing the quantization (6) from the requirement that the complex wave function $\Psi_{GL}=\left|\Psi_{GL}\right|\exp i\varphi$ must be single-valued at any point in the superconductor. According to this requirement, the phase φ must change by integral multiples 2π following a complete turn along a path *l* of integration $\oint_{l}dl\nabla \varphi =n2\pi$ [8]. Therefore, the relation
(18)$$\mu_{0}\oint_{l}dl\lambda_{L}^{2}j+\Phi =n\Phi_{0}\;$$
between the current density *j* along any closed path *l* and a magnetic flux $\oint_{l}dlA=\Phi$ inside this path must be valid. The expression deduced from the wave function $\Psi_{GL}=\left|\Psi_{GL}\right|\exp i\varphi$ in 1941 [59] allows us not only to explain the Meissner effect, but also to predict the quantization of a magnetic flux observed first in 1961 at measurement of a superconducting cylinder with thick wall $w\gg \lambda_{L}$ [4,5] and the quantization of the persistent current (9) observed first in 1962 at measurement of a superconducting cylinder with a thin wall $w\ll \lambda_{L}$ [6].

The current density has the same value along the wall in the case of weak screening at $w\ll \lambda_{L}$ according to (18), and therefore this expression gives the expression (9) for the persistent current. In the opposite case of strong screening at $w\gg \lambda_{L}$, there is a contour inside the thick wall along which the current density is zero j=0 and therefore $\Phi =n\Phi_{0}$ according to (18). The quantum number *n* can be non-zero if only a singularity of the wave function $\Psi_{GL}=\left|\Psi_{GL}\right|\exp i\varphi$ is inside the closed path *l*. A hole in the superconducting cylinder [4,5] and the Abrikosov vortex [7] are such singularities. The integral $\oint_{l}dl\nabla \varphi =n2\pi =0$ and the quantum number *n* must equal zero when the closed path *l* can be reduced to a point in the region inside *l* without singularity. Thus, the Meissner effect may be considered as a special case of the flux quantization $\Phi =n\Phi_{0}$ when the quantum number n=0 [40].

According to both the GL theory [1] and the BCS theory [2] the Meissner effect is observed due to the condensation of mobile charge carriers which are the Cooper pairs with the charge q=2e according to the BCS theory [2]. The GL theory [1] was created within the framework of equilibrium reversible thermodynamics and superconducting transition is the phase transition according to this theory. However, L.D. Landau and V.L. Ginzburg ignored the concern of W. H. Keesom that “*it is essential that the persistent currents have been annihilated before the material gets resistance, so that no Joule-heat is developed*” [14] in order the superconducting transition could be the phase transition. Therefore, the GL theory [1] is internally inconsistent. Other theorists, including J. Bardeen, L.N. Cooper and J.R. Schrieffer [2], were concerned with the problem of how electrons can become bosons. They have forgotten, perhaps because of the complexity of this problem, that Joule heating is an irreversible thermodynamic process. Therefore, the BCS theory [2] also contradicts the law of entropy increase.

### 5.3. The Meissner Effect Puzzle

Jorge Hirsch [9,10,11] has drawn reader’s attention to the internal inconsistency of the BCS theory [2] in order to prove the superiority of his theory of hole superconductivity [60] over the conventional theory. He has been trying for more than thirty years to convince the superconducting community that the conventional theory of superconductivity is inadequate and only his theory can describe superconducting phenomena without contradictions, see his numerous publications on the website https://jorge.physics.ucsd.edu/jh.html (accessed on 29 December 2021).

For a long time, he did not succeed. Hirsch compared the attitude of most physicists to the conventional theory of superconductivity with the attitude of the characters in Andersen’s fairy tale ’The Emperor’s New Clothes’ to the Emperor’s new clothes in the publication [61]. He points out in particular that the conventional theory of superconductivity [1,2] cannot explain the Meissner effect: “*’Heaven help me’, thought smart students that couldn’t understand how BCS theory explains the Meissner effect. ’I can’t possibly see how momentum conservation is accounted for and Faraday’s law is not violated’. But they did not say so*” [61].

The conventional theory of superconductivity [1,2], following Landau [59], considers the Meissner effect as a special case of the quantization (6): the persistent currents, both (9) and (15) appear since superconducting state with zero electric current is forbidden. However, the theory [1,2] says nothing about the force that could accelerate the mobile charge carriers against the action of the Faraday electric field $E=-{\left(2\pi r\right)}^{-1}d\Phi /dt$.

Hirsch expressed astonishment that this puzzle is ignored: “*Strangely, the question of what is the ’force’ propelling the mobile charge carriers and the ions in the superconductor to move in direction opposite to the electromagnetic force in the Meissner effect was essentially never raised nor answered to my knowledge, except for the following instances: [62] (H. London states: “The generation of current in the part which becomes supraconductive takes place without any assistance of an electric field and is only due to forces which come from the decrease of the free energy caused by the phase transformation,” but does not discuss the nature of these forces) [42] (A.V. Nikulov introduces a ’quantum force’ to explain the Little-Parks effect in superconductors and proposes that it also explains the Meissner effect)*” [63].

I should note that the “quantum force” [42] was deduced from the conventional theory [1] and experimental results in order to explain why the persistent current can be observed at a non-zero resistance. The deduction of the quantum force [42] does not go beyond the GL theory [1] and therefore cannot claim to explain what is the ’force’ propelling the mobile charge carriers when the persistent current appears in superconducting ring (9) and because of the Meissner effect (15). Hirsch claims that his theory of superconductivity can explain not only what force creates the persistent current (15), but also how this current can be annihilated before the material gets resistance, as W. H. Keesom wanted [14].

Hirsch’s theory requires radial flow of charge during the transition between normal and superconducting states. The persistent screening current (15) appears and is annihilated under the action of the Lorenz force due to this radial flow of charge, according to Hirsch’s theory. This explanation has two fundamental flaws: (1) according to Hirsch’s theory, an electric field directed from the center to the edge of a bulk superconductor must exist in the superconducting state [63]; (2) Hirsch’s theory cannot explain the appearance of the persistent current (9) in a superconducting ring in which no radial flow of charge is possible in principle.

Hirsch agrees with the opinion of most physicists that the Meissner effect was able to turn the irreversible thermodynamic process into the reversible one [64]. His theory, as well as the conventional theory of superconductivity [1,2], was created within the framework of equilibrium reversible thermodynamics. Therefore, Hirsch ignores the experimental evidences that the persistent currents are not annihilated not only before, but even after the material gets resistance [35,36] and that the persistent currents are observed even in normal metal rings [37,38]. He also ignores the obvious mistakes and contradictions that physicists had to make after 1933 in order to preserve their faith in the law of entropy increase.

### 5.4. Contradiction with Elementary Logic

The radical change of opinion about superconducting transition after 1933 fits with the law of entropy increase, but contradicts elementary logic: the process of the energy (16) dissipation of the electric current (15) in the normal state cannot depend on how this current appeared in the superconducting state. W.H. Keesom wrote in 1934: “*In passing (in the beginning of process 3) the threshold value curve, the electromotive force on the solenoid that maintains the magnetic field must do an amount of work equal to twice the energy of the field that comes into existence in the metal. Till now we imagined that the surplus work served to deliver the Joule-heat developed by the persistent currents the metal getting resistance while passing to the non-supraconductive condition. As, however, the conception of Joule-heat can rather difficulty be reconciled with reversibility we think now that there must be going on another process that absorbs energy*” [14].

It is extremely strange that even W.H. Keesom could not understand that the surplus work can increase only the Joule heat, despite that “*the conception of Joule-heat can rather difficulty be reconciled with reversibility*”. W.H. Keesom was understanding that the power source of the solenoid should perform the work
(19)$$A=\int_{t}dtIV=\int_{t}dtId\Phi /dt=I\Phi =\pi R^{2}\mu_{0}H^{2}\;$$
per unit length of the superconducting cylinder at its transition in the normal state in order to maintain the external magnetic field H=I. Half of this work is spent on creating the energy of the magnetic field $VBH/2=V\mu_{0}H^{2}/2$ inside the volume $V=L\pi R^{2}$ of the cylinder. It should be emphasized that the energy of the magnetic field VBH/2 is zero in the superconducting state since the magnetic flux density B=0. W.H. Keesom and other physicists did not doubt before 1933 that the surplus work, the second half of the work (19), generates Joule heat as well as no physicist can doubt that the second part of the work (19) generates Joule heat when a rapid change in the magnetic field from 0 to H=I induces the screening current in a normal metal cylinder.

W.H. Keesom and other physicists did not take into account, because of their faith in the law of entropy increase, that superconducting state with the persistent current (15) can be achieved in two ways. In the first way, considered above (see Section 4.4), the power source of the solenoid does not perform a work when the temperature decreases from $T>T_{c}$ to $T<T_{c}$ in the zero magnetic field and performs a small work equal to the kinetic energy (16) of the persistent current (15) when the magnetic field increases from 0 to H=I at $T<T_{c}$. The power source performs the main work (19) at the transition of the cylinder with a macroscopic radius to the normal state.

The kinetic energy (16) is $2\lambda_{L}/R\approx 0.0001$ times less than the energy of the magnetic field in the normal state $\mu_{0}H^{2}/2$ at typical values of the radius of the cylinder R≈1 mm and the London penetration depth $\lambda_{L}\approx 50$ nm = 0.00005 mm. Half of the work (19) returns to the power source when the magnetic field decreases from H=I to 0 at $T>T_{c}$. Thus, the power source in one cycle performs the work equal to the sum $\mu_{0}H^{2}\lambda_{L}\pi RL+\mu_{0}H^{2}/2$ of the kinetic energy (16) in the superconducting state and the energy of the magnetic field in the normal state. The cycle can be repeated many times. In which reservoir can the energy generated by this work accumulate, if it does not turn into Joule heat?

### 5.5. Contradiction with the Law of Energy Conservation

It is surprising that none of the physicists who believed that the superconducting transition became a phase transition after 1933 did not ask this obvious question. Even more surprising is the belief in the equality of the free energies at the critical magnetic field $H_{c}$ at which the superconducting cylinder goes into the normal state. According to the universal belief on which the theory of superconductivity [1,2] is based, superconducting transition is observed at $T=T_{c}$ since the free energy of superconducting state is smaller than the one of the normal state $F_{s0}<F_{n0}$ at $T<T_{c}$ and $F_{s0}>F_{n0}$ at $T>T_{c}$ in the absence of a magnetic field H=0. It was found already before 1933 that a bulk superconductor goes in the normal state at $T<T_{c}$ when external magnetic field reaches a critical value $H=H_{c}\left(T\right)$. Hirsch, following C. J. Gorter [65], writes in [64] that the free energy of superconducting and normal states should be equal $F_{s}=F_{n}$ at $H=H_{c}\left(T\right)$ in order the transition in the magnetic field could be reversible and wrote the equality
(20)$$F_{n0}\left(T\right)-F_{s0}\left(T\right)=V\frac{\mu_{0}H_{c}^{2}\left(T\right)}{2}\;$$
postulated by C.J. Gorter and others in 1933.

The equality (20) is written in almost all books on superconductivity [13,44] and underlies both the conventional theory of superconductivity [1,2] and Hirsch’s theory. However, this equality obviously contradicts the law of energy conservation. The equality (20) is deduced from the equality $F_{s}=F_{s0}+V\mu_{0}H_{c}^{2}\left(T\right)/2=F_{n}=F_{n0}$ of the free energy of superconducting $F_{s}$ and normal $F_{n}$ states when the external magnetic field equals the critical magnetic field $H=H_{c}\left(T\right)$. The equality of the free energy $F_{s}=F_{n}$ was postulated in 1933 in order superconducting transition observed at $H=H_{c}\left(T\right)$ could be considered as the phase transition. According to this equality the magnetic field $H=H_{c}\left(T\right)$ should increase the free energy of superconducting state $F_{s}=F_{s0}+V\mu_{0}H_{c}^{2}\left(T\right)/2$ rather than the normal state $F_{n}=F_{n0}$ since $F_{s0}<F_{n0}$ at $T<T_{c}$ and H=0. However, this postulate contradicts to the law of energy conservation since the power source of the solenoid performs in superconducting state the small work equal the kinetic energy (16), which is much smaller the work performed by the power source in order to create the energy $V\mu_{0}H_{c}^{2}\left(T\right)/2$ of magnetic field $H=H_{c}\left(T\right)$ in the normal state.

Hirsch seems to think that the energy $V\mu_{0}H_{c}^{2}\left(T\right)/2$ on the right side of the Equation (Equation 20) is the kinetic energy of the surface current (15) in the superconducting state. In order to convince himself and others of this, Hirsch deduced the equality (B13) from the equality (B7) in [64]. According to the equality (B7) and (B13) the energy of the magnetic field in the volume *V* of a bulk cylinder with a macroscopic radius $R\gg \lambda_{L}$ in the normal state $V\mu_{0}H^{2}/2$ equals the density of the kinetic energy $\varepsilon =n_{s}mv^{2}/2=\mu_{0}\lambda_{L}^{2}j^{2}/2=\mu_{0}H^{2}/2$ of the persistent current in the superconducting state at r=R. But the energy cannot be equal the energy density in principle. Hirsch made obvious mathematical mistake in [64] in order to convince himself and others that the equality (20) is correct and superconducting transition is the phase transition.

The mathematical mistake made by Hirsch in [64] has provoked a delusion of H. Koizumi [66], who thinks that the kinetic energy of the persistent current in the superconducting state equals the energy of the magnetic field in the normal state because of his belief that Hirsch deduced correctly the equality (B13) from the equality (B7) in the paper [64]. Therefore, he writes that “*The reversible process implies $\Delta F_{m}+\Delta F_{k}=0$*” [66], i.e., that the kinetic energy $F_{k}$, see (2) in [66], is transformed in the energy of the magnetic field $F_{m}$ see(1) in [66], at the transition of a bulk superconductor to the normal state. Such a transition is certainly reversible. Not only an irreversible increase in entropy, which occurs during Joule heating, but even a reversible increase in entropy, which occurs during the phase transition of the first kind, is absent when the kinetic energy is transformed in the energy of the magnetic field. However, such a transformation cannot be possible since the kinetic energy in superconducting state (16) is $2\lambda_{L}/R$ times less than the energy of the magnetic field in the normal state [67].

Unfortunately, not only does H. Koizumi [66] not know how a reversible thermodynamic transition differs from an irreversible transition. Therefore, this difference is explained in the Comment [67] on the article [66]. The blind faith in the law of entropy increase provoked not only the contradiction with the law of energy conservation and mathematical mistake, but also profanation of physics. Only this blind faith could force physicists to forget that the energy density VBH/2 of the magnetic field *H* in the volume *V* of a bulk superconductor is zero when the magnetic flux density is B=0 zero rather than when $B=\mu_{0}H$.

### 5.6. No Work Can Be Performed during Any Phase Transition

The blind faith has forced physicists to also forget that the change in the magnetic flux Φ in a bulk superconductor during the superconducting transition induces the Faraday electromotive force −dΦ/dt that can perform work on external bodies. Or could physicists have forgotten that no work can be performed during the phase transition when they concluded that the Meissner effect was able to turn the irreversible transition into the phase transition? No work can be performed because of the equality of the free energy $F_{s}=F_{n}$ postulated at any phase transition. The free energy by definition F=H−ST is a part of the internal energy *H* that is able to perform work in the absence of a temperature difference. Therefore, the free energy should decrease when a work *A* is perform during the superconducting transition and the equality $F_{s}=F_{n}$ cannot be possible according to the law of energy conservation.

A visual example of the work performed during the superconducting transition is the levitation of a small magnet over the surface of a superconductor. The potential energy of the magnet with a mass *m* in the gravitational field increases on the value gmh when the magnet lifts to a height *h* due to the transition of a superconductor into the superconducting state. This well-known example is enough to refute the belief of several generations of physicists that the superconducting transition is the phase transition. The free energy of the superconductor must be reduced on the value of the work A=gmh, performed in order to increase the potential energy of the magnet. Here, we may recall that Smoluchowski argued in 1914 [16] the impossibility of a useful perpetuum mobile, proving that the potential energy of a particle at the Brownian motion cannot increase by an amount much greater than the energy of thermal fluctuations $k_{B}T$. Now, we see that the potential energy of the magnet increases by an amount much greater than this energy $gmh\gg k_{B}T$ due to the Meissner effect.

## 6. Conclusions

The problem of violation of the law of entropy increase has both fundamental and practical importance. Elliott Lieb and Jakob Yngvason wrote that this “*law has caught the attention of poets and philosophers and has been called the greatest scientific achievement of the nineteenth century. Engels disliked it, for it supported opposition to dialectical materialism, while Pope Pius XII regarded it as proving the existence of a higher being*” [17]. The history of this law shows that our ideas about Nature are based not only on experience and logic, but also on faith. The supporters of the thermodynamic-energy worldview denied at the end of the nineteenth century the existence of atoms and their perpetual thermal motion because of their faith in this law.

The experience of the Brownian motion and other fluctuation phenomena, investigated at the beginning of the twentieth century, convinced most of the supporters of this worldview in the existence of the perpetual thermal motion of atoms and other particles. However, the faith of twentieth-century physicists turned out to be more blind than the faith of nineteenth-century scientists. Only the blind faith could force W.H. Keesom and others to postulate the false equality of free energies (20), which became the basis for understanding superconductivity, despite the fact that they knew that the power source of the solenoid performs work during the transition of the bulk superconductor to the normal state, and not during the reverse transition.

The main psychological argument of believers in the law of entropy increase is the assertion that no one has managed to make a perpetuum mobile for several centuries. However, for example, a nuclear reactor also could not be made for many centuries. A nuclear reactor and a perpetuum mobile differ first of all in the time of the appearance of the task of their creation. The task of creating a nuclear reactor could have arisen only in the twentieth century, when nuclear fission was discovered, while from time immemorial, everyone could be convinced by experience that the cart would stop if it was not pushed. The laziness inherent in people made them dream about a perpetuum mobile. However, not every dream comes true right away. Now first of all, the belief in the impossibility of a perpetuum mobile prevents the dream of a perpetuum mobile from coming true. It is possible the perpetuum mobile would have been created before the nuclear reactor without this belief.

Most, but not all, scientists believe in the impossibility of a perpetuum mobile. There are published articles [12,68,69,70,71,72] and even books [73] in which this belief is questioned. Several conferences devoted to the problem of the universality of the law of entropy increase have been held, the results of which have been published [74,75,76]. The Special Issue of Entropy [30] was dedicated to this problem. P.D. Keefe considered in the articles [77,78,79,80] a contradiction of the Meissner effect with the law of entropy increase. Nevertheless, the law of entropy increase remains a matter of faith rather than understanding for most scientists.

Of course, not all doubts in this law are well-founded and even scientific. However, these doubts should be refuted by scientific arguments, and not banned from publication only on the basis of the belief of the majority in the impossibility of a perpetuum mobile. Most scientists have to understand that the supreme position of the second law of thermodynamics has no scientific justification. This position is determined rather by the psychology of scientists who find it difficult to recognize that the centuries-old belief of most scientists can be false. Possibility of violation of the second law of thermodynamics should be openly discussed in scientific journals.

This work was made in the framework of State Task No 075-00355-21-00.

## Data Availability

The data presented in this study are available in article here.

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
