# Peer review of "The Law of Entropy Increase and the Meissner Effect"

_entropy, 2022, doi:10.3390/e24010083_

Round 1
Reviewer 1 Report
The paper discusses the possibility of entropy increase violation, in particular from the point of view of superconductivity. It is a well written piece, which is pleasant to read even to a non specialist like me. I will not judge the relevance of the debate about possible violation of the second law of thermodynamics; let me just recall that every empirical fact contradicting a theory is worth being taken into consideration, in the name of the refutability/falsifiability principle. I will therefore suggest only minor modifications regarding the form and presentation, and leave to other specialist reviewers the decision regarding the scientific value of the authors'arguments.
- I think the abstract should present clearly the paper's main purposes; under its present form one does not understand the paper's motivations and main conclusions. Same remark holds for the introduction.
- Regarding the efficiency, I would recommend to use the greek letter $\eta$ which is a more commonly used notation, instead of "eff".
- Also, I believe that it is misleading to present eq. (1) as Carnot's principle. Better formulation is Carnot's theorem, which is of course itself a consequence of Carnot's principle (second law of thermondynamics).
- First sentence in p. 6: "An useful work cannot obtained from the Brownian motion because of its disorder". It seems a verb is missing.
- Section 4.2: when it comes to the BCS theory, I am suprised that the standard terminology of "Cooper pairs" is not introduced in the discussion. According to my knowledge of the subject (which is remanent of my Physics studies more than 10 years ago!), the most commonly accepted cause of the Meissner effect is the condensation of Cooper pairs, which behave essentially like bosons (even if electrons are fermions). I think it would be worth introducing both these facts in a more pedagogical perspective, so that the reader can understand better the author's criticism.
- Last, I would recommend to shorten the conclusion, and that the author avoids unnecessary complaints about the censorship he claims to be victim of. I do not deny the possible existence of existence of censorship if some scholarly journals, but the comparison between his situation and Boltzmann's is, to say the least, risky.
Author Response
I thank Reviewer 1 for his constructive and useful comments. I have revised the manuscript in accordance with his recommendations.
I have rewritten the abstract and the introduction in accordance with the recommendation of Reviewer 1.
The efficiency has been denoted with the greek letter $\eta$ instead of "eff".
The equation (1) has been presented as the Carnot equation instead of the Carnot principle.
The missing verb “be” is added to the first sentence on page 6: "A useful work cannot be obtained from the Brownian motion because of its disorder"
I have replaced the term "superconducting pairs" with the more standard term "Cooper pairs".
It is correct that the most commonly accepted cause of the Meissner effect is the condensation of Cooper pairs. But I think it is important to pay attention to the fact that L.D. Landau was the first who used the condensation to explain the Meissner effect in 1941, at the end of his article, L.D. Landau, Theory of Superfluidity of Helium II. Zh.Eksp.Teor.Fiz. 11, 592-623 (1941). L.D. Landau described the superconducting condensation with the help of a wave function. This wave function written in 1941 became the basis of the famous Ginzburg-Landau theory published in 1950. Of course, Landau's assumption about electron condensation was illegal from the point of view of the fundamentals of quantum mechanics. But the wave function written by L.D. Landau in 1941 allowed not only to explain the Meissner effect but also to predict the quantization of magnetic flux observed at the first time in 1961.
I have shortened the conclusion in accordance with the recommendation of Reviewer 1.
Reviewer 2 Report
Authors refute the modern belief in the second law of thermodynamics that entropy increases. My main concern is it is not clear if there is no (physical) connection between examples that authors have discussed such as Meissner effect or average Langevin force value being nonzero, and surroundings.
In a simple example of heat engine as discussed by authors, useful work is done by heat engine only when there is a temperature difference. The temperature difference connects heat engine with surroundings, allowing transfer of heat to surroundings. There is an increase in the entropy in surroundings, resulting in a net increase in entropy of the system (heat engine + surroundings), although there is a decrease in the entropy of engine, which is responsible for the directional movement by the engine.
I think authors have to clarify if the examples discussed are isolated thermodynamically, such as at the same temperatures.
Authors must give the names of the authors from citations in their manuscript, which are just indicated simply by 'authors'.
This manuscript needs scientific language editing.
Author Response
I thank the Reviewer for his comments and expression of his main concern.
Because of this concern I have added in the end of section 4.5 the text:
“According to the Carnot equation (1) a temperature difference is needed in order to create the directional movement with the help of a heat engine. But the persistent current $\overline{I_{p}}(\Phi )$ is already the directional movement observed \cite{Moler2007,Letters2007,PC2009Science,Moler2009} at thermodynamic equilibrium in the absence of a temperature difference. The persistent voltage $V _{dc}(\Phi ) \propto \overline{I_{p}}(\Phi )$ observed on the halves of the asymmetric rings can be used to perform a useful work in the absence of a temperature difference due to a non-zero value of its power $P_{dc} = V _{dc}\overline{I_{p}} = V _{dc}^{2}/R$ at the zero frequency $\omega = 0$. The Nyquist noise cannot be used for a useful work without a temperature difference since each element of the electrical circuit has the same power $P_{Nyq} = k_{B}T\Delta \omega$ in any frequency range $\Delta \omega$.
Therefore, the power $P_{Nyq} = k_{B}T\Delta \omega$ will always flow from a hot body to a cold body, in accordance with the law of entropy increase, no matter what frequency filters are used. The DC power can be transferred from a cold body to a hot body when the filters pass only the DC current. The DC voltage $V _{dc}(\Phi ) \propto \overline{I_{p}}(\Phi )$ induced on 1080 asymmetric rings at the temperature $T \approx 1 \ K$ is observed at the room temperature $T \approx 300 \ K$ in the experimental work \cite{Physica2019}. The DC power $P_{dc} = V _{dc}\overline{I_{p}} = V _{dc}^{2}/R$ is observed in \cite{Physica2019} since the directed electric current $\overline{I_{p}}$ flows against the directed electric field $E = -\nabla V_{dc}$ in one of the halves of asymmetric superconducting ring due to the violation of the assumption of molecular disorder in the quantum phenomenon of the persistent current”.
I have give the names of the authors from citations in the manuscript in accordance with the recommendation of the Reviewer.
The scientific language of the manuscript was edited.
Round 2
Reviewer 2 Report
I am satisfied with the revisions.